# Dietary Phytochemicals as Potential Chemopreventive Agents against Tobacco-Induced Lung Carcinogenesis

**DOI:** 10.3390/nu15030491

**Published:** 2023-01-17

**Authors:** Yan Ding, Ruilin Hou, Jianqiang Yu, Chengguo Xing, Chunlin Zhuang, Zhuo Qu

**Affiliations:** 1College of Pharmacy, Ningxia Medical University, 1160 Shengli Street, Yinchuan 750004, China; 2Department of Medicinal Chemistry, University of Florida, 1345 Center Drive, Gainesville, FL 32610, USA; 3School of Pharmacy, Second Military Medical University, 325 Guohe Road, Shanghai 200433, China

**Keywords:** dietary phytochemicals, lung cancer, tobacco carcinogens, chemoprevention

## Abstract

Lung cancer is the second most common cancer in the world. Cigarette smoking is strongly connected with lung cancer. Benzo[a]pyrene (BaP) and 4-(N-methyl-N-nitrosamine)-1-(3-pyridyl)-butanone (NNK) are the main carcinogens in cigarette smoking. Evidence has supported the correlation between these two carcinogens and lung cancer. Epidemiology analysis suggests that lung cancer can be effectively prevented through daily diet adjustments. This review aims to summarize the studies published in the past 20 years exploring dietary phytochemicals using Google Scholar, PubMed, and Web of Science databases. Dietary phytochemicals mainly include medicinal plants, beverages, fruits, vegetables, spices, etc. Moreover, the perspectives on the challenges and future directions of dietary phytochemicals for lung cancer chemoprevention will be provided. Taken together, treatment based on the consumption of dietary phytochemicals for lung cancer chemoprevention will produce more positive outcomes in the future and offer the possibility of reducing cancer risk in society.

## 1. Introduction

Lung cancer is the second most common cancer in the world. Its mortality in the past four decades was 59.0/100,000 person-years. Lung cancer was the leading cause of cancer death in 2020, representing approximately one in 10 (11.4%) cancers diagnosed and one in 5 (18.0%) deaths [1]. Meanwhile, the 5-year survival rate was less than 21% [2]. In 2022, approximately 350 Americans die from lung cancer every day [3]. In China, lung cancer is the most common cancer among males (50.04/100,000) and the second in females(23.63/100,000) [4]. The major types of lung cancer have been divided into small-cell lung carcinoma, and non-small cell lung cancer (NSCLC). NSCLC includes adenocarcinoma, squamous cell carcinoma, small cell, and so on. The occurrence of lung carcinogenesis is multifactorial. However, in most cases, it is due to familial inheritance and the epigenetic damage caused by long-term exposure to carcinogens. Besides, gene mutation could lead to the docking of downstream molecular signals and related protein changes and eventually lead to the occurrence of lung tumors. For example, EGFR (epidermal growth factor receptor, which plays an important role in the cancer process) activated mutations lead to constitutive tyrosine kinase activation and oncogenic transformation of lung epithelial cells [5]. Currently, there are many treatment strategies for lung cancer. Nowadays, immunotherapy, especially for programmed death-1 (PD-1)/PD-L1 blockade immunotherapy, as monotherapy, or combined with chemotherapy is the standard of care. However, for patients without an actionable driver mutation and when targeted therapies are not available, chemotherapy (in combination with radiotherapy) has been the standard of care, especially for advanced NSCLC [6]. Surgical resection is preferred for stages I through IIIA NSCLC. About 40% of the new diagnoses of cancer are in an advanced stage [7]. Therefore, the prevention of lung cancer needs more attention because of the poor survival rate.

According to AACR CANCER PROGRESS REPORT 2022, tobacco use is one of the main preventable causes of cancer. Smoking contributed to more than 64 percent of deaths of tracheal, bronchus, and lung cancers. For lung cancer, it is estimated that 90% of lung carcinomas are directly due to smoking and smoking-derived carcinogens. Consequently, the best means for preventing lung carcinoma is to avoid exposure to cigarette smoke [8]. It is worth mentioning that not all kinds of lung cancer are caused by smoking, especially for some Asian women. That is because of the activated mutation of EGFR [9]. Tobacco smoke consists mainly of nicotine, benzo[a]pyrene (BaP) and 4-(N-methyl-N-nitrosamine)-1-(3-pyridyl)-butanone (NNK) (for chemical structures, see Figure 1), which are highly toxic and can lead to health problems such as lung cancer. At present nicotine has not been identified as a carcinogen. However, nicotine addiction is a major contributor to persistent exposure to other carcinogens in the smoker. Moreover, nicotine has the effect of inducing chemotherapeutic drug resistance and activating nicotinic acetylcholine receptors to accelerate the development of cancer, making the treatment of patients with lung cancer less effective than expected [10]. BaP is one of the most studied polycyclic aromatic hydrocarbons, which has potent carcinogenicity, teratogenicity, and mutagenicity. Exposure to BaP has three significant routes, which are skin exposure, dietary intake, and inhalation. BaP is hydrophobic and readily crosses the cell membrane and binds to receptors. For instance, BaP could activate and up-regulate the expression of aryl hydrocarbon receptor (AhR) in lung adenocarcinoma [11]. It forms an intracellular complex with AhR that induces the expression of cytochrome P450 to stimulate its metabolism [12]. Benzo[a]pyrene-7,8-diol-9,10-epoxide (BPDE), a metabolite of BaP, could produce DNA damage and form BPDE-adducts [13]. Smokers tend to develop lung cancer when they are exposed to NNK. For the smoker, NNK absorption is regulated mainly by direct inhalation of mainstream smoke [14]. Non-smokers could be exposed to NNK by second-hand smoke and side-stream smoke from the smoldering tobacco product, and even dermally via NNK-contaminated surfaces [15]. Induction of lung cancer by NNK is mainly achieved through the following pathways: (1) Induces gene mutation; NNK is metabolized to NNAL by the action of CYP450 enzymes to induce DNA damage; (2) Activates α7 nicotine acetylcholine receptor and promotes cell proliferation, metastasis, angiogenesis, and reduce apoptosis in LC cells; (3) Regulates gene expression through DNA-methyltransferase1 (DNMT-1, a key enzyme mediating DNA methylation, is known to be elevated in various cancers) mediated epigenetic modifications; (4) Activates TxAS/TxA_2_/TP/PI3K/AKT/CREB and β-AR/arachidonic acid signaling pathways; (5) Induces immune suppression [16]. Therefore, it can be seen that tobacco carcinogens are one of the key factors to induce the occurrence and development of lung cancer.

Chemoprevention is becoming more and more important in lung cancer treatment. Epidemiology analysis suggests that lung cancer can be effectively prevented through daily diet adjustments [17]. The conventional treatments of lung cancer, such as surgery, radiotherapy, and chemotherapy, can reduce tumor sizes and inhibit the cancer cell growth but cannot cause complete regression. Meanwhile, immunosuppression and the effects of cancer itself lead to low life expectancy after these treatments [18]. Chemoprevention by natural or synthetic chemicals through reducing the exposure to tobacco carcinogens could lower the risk [19]. Nature offers many safe, effective, and affordable products, usually called “phytochemicals” for various diseases [20]. For example, researchers found a significant correlation between cruciferous vegetables and organic sulfur compounds-rich species such as garlic and reduced lung cancer risk [21,22]. In addition, vitamin-rich fruits such as oranges have potential chemoprotective effects on lung cancer [23,24]. Besides, some dietary natural products such as particular food groups and nutrients have attracted much attention due to their cancer prevention in the past decades. It is worth mentioning that AACR CANCER PROGRESS REPORT 2022 emphasizes that eating at least 30 g fiber and at least 400 g of fruit and vegetables each day could decrease cancer risk in humans. Therefore, the consumption of dietary natural products to prevent lung cancer seems to be an effective strategy. Moreover, lung cancer prevention, especially for dietary chemoprevention, has been paid more and more attention in recent years. With the aim of studying chemopreventive strategies, a wide variety of murine model systems have been developed. Among several strains of mice tested, the A/J mouse strain is very susceptible to developing lung tumorigenesis, with the incidence of spontaneous and induced lung tumors at the highest rate of 61% [25,26]. It has been reported that A/J mice are more sensitive to tobacco carcinogens than other species of mice. In addition, A/J mice develop spontaneous tumors even in the absence of a carcinogen stimulus [27]. Thus, the A/J mouse model of lung cancer induced by tobacco carcinogens such as NNK or BaP is widely used in lung cancer chemoprevention [28].

Based on the above, this review aims to analyze those research studies focusing on the chemopreventive effects of dietary phytochemicals on NNK or BaP-induced lung carcinogenesis (see Figure 1 and Table 1) and to summarize the clinical trials of phytochemicals (see Table 2). We also discuss the chemopreventive mechanisms of listed dietary natural resources on lung cancer. To achieve this goal, we first obtained the related studies published in the last 20 years, mainly using Google Scholar, PubMed, and Web of Science databases. The search words included “Phytochemicals” OR “fruits and vegetables” OR “beverages” OR “species” AND “Lung cancer” AND “cigarette smoking” AND “NNK” OR “BaP” OR “nicotine”. The search is limited to in vivo studies.

## 2. Chemoprevention of Lung Cancer by Dietary Phytochemicals

Dietary natural products such as medicine plants, beverages, fruits, vegetables, spices and other plants possess anti-inflammatory, anti-oxidative, anti-proliferative, and cancer inhibitory activities, which have attracted much attention in the past decades. This section focuses on lung cancer chemoprevention properties of these dietary resources and the active compounds in vivo (chemical structures see Table 3). 

### 2.1. Medicine Plants

Medicinal plants have been used in many countries as a source of primary healthcare since ancient times because of their effectiveness and low cost, as well as lesser side effects than many modern treatments [104]. Some traditional Chinese medicines are also considered as “drug homologous foods” for developing health products, such as glycyrrhiza, haskap berry, emblica, and psoralea. These “drug homologous foods” are effective in lung cancer prevention. 

#### 2.1.1. Glycyrrhiza

In both traditional Ayurvedic and Chinese medicine, *Glycyrrhiza glabra* L. is commonly referred to as Mulaithi and Yashtimadu, and is used to treat a variety of illnesses. Abundant components are present in *Glycyrrhiza glabra* L., such as flavanones, chalcones, isoflavans, isoflavenes, flavones, and isoflavones. The flavonoid lipochalcone A (LicA), isolated from Glycyrrhiza uralensis Fisch, is a potent pharmacological agent with a wide spectrum of activities. Among NNK-induced A/J mice, LicA can reverse the expression of NNK-induced miRNA miR-144-3P, miR-20a-5P, miR-29c-3P, let-7d-3p and miR-328-3p, to play a chemical prevention role in lung cancer [29]. Glycyrrhizin, a triterpenoid saponin isolated from glycyrrhiza, reversed high level of High Mobility Group Box 1 (HMGB1), which is a protein that regulates nucleosome formation and gene transcription. In addition, glycyrrhizin decreased the activity of JAk/KSTAT signing pathway in PDX mice [30]. Glycyrrhizin plus cisplatin or glycyrrhizin alone all reduced the expression of thromboxane synthase (TxAs) and PCNA in nude mice [31,32]. Proliferative cell nuclear antigen (PCNA) is necessary for DNA synthesis and DNA repair. In conclusion, as potential phytochemical *Glycyrrhiza glabra L*. has the potential to be a health product. The mechanism of licorice and its extracts or monomers in treating diseases is not completely clear, and it is too early for licorice to enter the clinical frontline as a low toxic, effective and controllable drug. However, as a health product or placebo, it has great commercial potential.

#### 2.1.2. Haskap Berry

*Lonicera caerulea* L., also known as haskap, is a fruit, commonly planted in Asia, Canada and Eastern Europe. It is also well known by the Japanese aborigines as the “elixir of life”. Haskap berry is rich in ingredients of phenolic acids and flavonoids, especially anthocyanins. Increasing evidence has suggested that haskap berry can prevent the lung tumorigenesis. BEAS-2B cells pre-treatment with the anthocyanin-rich haskap berry extracts significantly reduced 4-[(acetoxymethyl) nitrosamino]-1-(3-pyridyl)-1-butanone (NNKOAc), a precursor of NNK, which could induce DNA damage [105]. About 79–92% of the anthocyanin content of haskap berries is composed of cyanidin-3-O-glucoside (C3G). C3G shows significant antioxidant, cardio-protective, anti-inflammatory, neuroprotective, anticancer, and anti-diabetic properties [106]. It is documented that NNK-induced A/J mice treated with C3G could markedly decrease the expression of PCNA and Ki-67 in lung tissues [33]. In an experiment of C3G plus 5-FU (a commonly used basic chemotherapy drug) in nude mice against lung cancer, the inflammatory cytokine, IL-1β, IL-6 and TNF-α, COX-2, NF-kpa and PCNA was decreased [34]. Due to this, haskap berry, especially its major component C3G, may be a promising dietary supplement or nutraceutical for suppressing lung cancer development among high-risk populations. However, this is limited to animal experiments and is still a long way from the clinic. Its future use may still be limited.

#### 2.1.3. Emblic

*Phyllanthus emblica* L. (PEL), also named emblic, is a medical plant, which has been used extensively in Asia. Investigations have shown that PEL possesses a wide range of phytochemicals including tannins, polyphenols, gallic acid, flavonoids and vitamin C. Bioactivity studies revealed that PEL has a variety of biological activities, including anti-cancer. Wang and others found that the PEL extracts significantly reduced the number of BaP-induced A/J mice lung surface nodes. PEL extracts regulated the IL-1β/miR-i101/Lin28B signaling pathway for lung cancer prevention [35]. PEL exhibits anticancer activity against lung cancer, and is worth further study as a possible chemopreventive and anti-invasive agent.

#### 2.1.4. Psoralea

Psoralea, also named *Psoralea corylifolia* L., is a traditional medicine used in ancient Egypt and China for various ailments such as psoriasis, vitiligo, hair loss, impotence, osteoporosis, etc. Phytochemical studies showed that psoralea mainly contains coumarins, flavones, monoterpenes, chalcones, lipids, resins, and stigmasteroids. 8-methoxsalen (8-MOP) is a furocoumarinisolated from this plant. Laboratory studies showed its effectiveness in chemoprevention of lung cancer Regarding the chemopreventive properties, 8-MOP was demonstrated to inhibit the CYP2A5-mediated metabolic activation of NNK in the mouse lung, leading to the prevention of NNK-induced adenoma [36,37,38]. However, 8-MOP in the diet of post-initiation phase does not prevent mouse lung carcinogenesis [107]. This effect is also confirmed in C57BL/6 mice that are induced by NNK [39]. In conclusion, 8-MOP is a strong chemopreventive agent for NNK-induced lung tumorigenesis.

### 2.2. Beverages

#### 2.2.1. Tea

Tea (Camellia sinensis) is a popular beverage worldwide. Most of the tea is consumed in America and green tea is mainly consumed in Asian countries. Tea may help reduce the risk of cardiovascular disease, anti-hypertensive effects, body weight control, antibacterial activity, anti-fibrotic properties, solar ultra violet protection, and neuroprotective power. (-)-epigallocatechin-3-gallate (EGCG) and caffeine are main active components in tea. EGCG, the major catechin found in tea, has the potential to impact a variety of human diseases. A/J mice treatment with EGCG could weaken the elevation of DNMT1 and inhibit lung tumorigenesis triggered by NNK [40]. Tea, coffee, and soft drinks are the most important caffeine sources and energy beverages. Caffeine in tea plays an important role in preventing tumorigenesis. It is reported that administration of caffeine could inhibit the developmentof lung adenoma into adenocarcinoma in NNK-induced A/J mice [41]. Caffeine 680 ppm significantly lowered the incidence of lung tumors from 47% to 10% in F344 rats [42]. Natural polyphenols are organic chemicals thatcontain phenol units in their structures. Polymeric black tea polyphenols (PBPs) are the most abundant polyphenolic component in black tea. Several laboratory studies have confirmed the lung cancer preventive properties of PBPs. Administration of PBPs inhibited cell proliferation rate and the progression of adenoma to adenocarcinoma in adenomas of NNK-induced A/J female mice [43]. In addition, PBPs have chemopreventive effects in NNK- and BaP-induced A/J mice possibly via modulation of p38 and Akt to inhibit inflammation and cellular proliferation and to induce apoptosis [44]. In addition, PBPs decreased BPDE-DNA adducts and inhibited carcinogen-induced phase-I enzymes of lung lesions in A/J mice [45]. Polyphenon E is a green tea polyphenol preparation that contains 65% EGCG, significantly reducing the incidence and multiplicity of adenocarcinoma, similarly to PBPs [41]. Polyphenon E plus atorvastatin, or polyphenon E alone both lowered tumor incidence and tumor burden of lung carcinogens. However, atorvastatin alone only decreased tumor burden not tumor multiplicity [46]. This means the effects of lung cancer prevention maybe contributed to by polyphenon E [48]. NNK-treated mice using green tea containing drinking water (2%) significantly decreased lung tumors per mouse and inhibited the increase of 8-OH-dGuo levels in mouse lung DNA [49]. 0.3% green tea extracts also reduced tumor nodes and decreased PD-L1 positive cells in lung tumors of A/J mice [50]. White tea extracts also significantly lowered the level of ROS and lipid peroxidation (LPO) [51]. Catechin hydrate is a mixture of gallocatechingallate and epigallocatechingallate. Compared with BaP group, catechin hydrate + BaP group reduced Bcl-2, Bax and Capase-3 expression in lung tissues [47]. Tea, especially for PBPs, maybe a promising product in the area of health. As one of the beverages that people have the longest contact with in their life, tea has relatively more preventive effects on lung cancer among all the plants that have been studied. However, the exact preventive mechanism of tea is still unclear, but as a food, it is worth more use.

#### 2.2.2. Kava

##### Kava

Kava is the extract of the roots of *Piper methysticum*. It has been consumed safely as a beverage in the South Pacific islands for centuries. According to some reports, it has anti-cancer and anti-anxiety effects. Our group has carried out continuous chemoprevention studies on kava for lung cancer. The results displayed that dietary kava led to statistically significant lower lung tumor multiplicities in NNK plus BaP-treated A/J mice [55]. It is worth mentioning that diets containing kava reduced lung adenomas multiplicity by approximately 99%. Furthermore, daily gavage of kava blocked lung adenoma formation by decreasing DNA damage in lung tissues of O^6^-methylguanine (O^6^-mG) [52]. Moreover, mice that were fed diets containing kava suppressed lung tumorigenesis in A/J mice via inhibition of cell proliferation, enhancing apoptosis, and also inhibiting the activation of NF-κB [53]. In addition, kava and its component methysticin suppressed NF-κB activation in lung adenoma tissues with minimum toxicity [56]. However, a three-day kava pretreatment potentiated acetaminophen-induced hepatotoxicity increasing severity of liver lesions, resulting from the cytotoxicity of chalcone compounds in kava [54,57]. In a clinical trial to evaluate the effects of kava on NNK metabolism, kava increased urinary excretion of total NNAL and reduced urinary 3-methyladenine (3-mA) [101]. 

##### Dihydromethysticin

Dihydromethysticin (DHM) is one of the active ingredients in kava. DHM had a dose-dependent chemical prevention effect on NNK-induced lung adenoma. DHM could reduce the formation of O^6^-mG and increase urinary excretion of NNK detoxification metabolites [58]. However, another study found that DHM does not affect NNAL formation and does not inhibit NNAL bioactivation. However, DHM remarkably increased O-glucuronidated NNAL in an NNK-induced model [59]. Synthetic DHM decreased adenoma multiplicity by 97% in NNK-induced A/J mice, which effects were equal to natural DHM, and it reduced DNA adducts in lung tissues [60]. The in vivo structure-activity relationship study found that the unnatural enantiomer of DHM was more potent than the natural enantiomer in NNK-induced A/J mice. More importantly, DHM analogues with high structural similarity show distinct efficacy in reducing NNK-induced DNA damage [61]. Moreover, it was determined that the methylenedioxy functional group in DHM that effectively reduced O^6^-mG adducts formation was critical to the chemopreventive activity and the lactone of DHM could be modified [61,62]. It was also found that DHM decreased the level of O^6^-mG in female C57BL/6 mice. However, in the backgrounds of Ahr^+/−^ and Ahr^−/−^, there is no difference inO^6^-mG, indicating that AhR is not the upstream target for DHM [64]. In a recent study, DHM was found to suppress the activation of protein A (PKA) pathway induced by NNK [63]. Kava and its compounds need more attention to explore their effects in lung cancer. Kava, especially DHM, plays an obvious role in the prevention of lung cancer in mice, but its role in the prevention of lung cancer in patients with a high risk of lung cancer needs to be further studied.

### 2.3. Fruits 

#### 2.3.1. Citrus

##### 5-Demethylnobiletin

Citrus is the most eaten fruit. This fruit is thought to decrease the risk of degenerative diseases and cancer. Active phytochemical constituents of this plant have been revealed so far, namely, limonene, β-caryophyllene, geranial edulinine, ribalinine, 5-demethylnobiletin, and nobiletin. 5-Demethylnobiletin is a unique citrus flavonoid with various beneficial bioactivities. Song et al. first confirmed that dietary intake of 5-demethylnobiletin remarkably inhibited tumorigenesis in NNK-induced mice, and this effect possibly was due to the metabolites of 5-demethylnobiletin in lung tissues [65]. 

##### Nobiletin

Nobiletin, a major component of citrus polymethoxy flavones, has anti-cancer effects [108]. Nobiletin inhibited CYP enzymes involved in the metabolic activation for preventing lung cancer [66]. In NNK-treated mice, oral nobiletin significantly suppressed lung tumorigenesis and decreased tumor volume [67].

##### Other Compounds

Limonin is a dietary phytochemical, which has been reported to possess many biological activities. Limonin supplements significantly decreased lipid peroxidation, serum marker enzymes and inflammatory cytokine levels in BaP-treated mice, and increased the nonenzymatic and enzymatic antioxidant levels, showing an effective lung cancer prevention [68]. β-Cryptoxanthin (BCX) is an oxygenated carotenoid and abundant in oranges, peaches, and pumpkins. BCX inhibited lung tumorigenesis through the down-regulation of α7-nAChR/PI3K signaling in NNK-induced A/J mice [69]. In addition, in the same lung cancer model, dietary intake of BCX restored the expressions of lung SIRT1, p53, and RAR-β, and decreased the levels of phosphorylate AKT and lung IL-6 mRNA to increase the survival probability in NNK-induced A/J mice [70]. Hesperetin and naringenin are the most abundant flavonoids in citrus fruits and possess various effects on different diseases. Hesperetin possesses a wide array of pharmacological effects which include cancer prevention [109]. Both hesperetin and naringenin effectively suppressed lung carcinoma and the associated preneoplastic lesions in a BaP-induced model by decreasing the expression of NF-kB, PCNA and CYP1A1 [71,110]. Moreover, hesperidin attenuates mitochondrial dysfunction during BaP-induced lung carcinogenesis in mice [72]. Citrus is one of the most commonly eaten fruits in daily life. According to the current literature the role of lung cancer prevention is currently documented only in animal experiments, and there is no literature to support the effects of low doses. Therefore, in life, for the purpose of preventing lung cancer, it can be appropriate to eat more of this fruit.

#### 2.3.2. Tomatoes

Tomatoes are a good source of health-promoting phytochemicals, including phenolic compounds, lycopene, vitamins and glycoalkaloids. Lycopene is a naturally occurring carotenoid in tomatoes, tomato products and other red fruits as well as vegetables. Among smokers, dietary intake of lycopene has a strong connection with the risk of lung cancer. Rakic et al. found that dietary lycopene effectively inhibited chronic bronchitis and preneoplastic lesions in a ferret model of cigarette smoke (CS)-induced lung cancer. Moreover, lycopene feeding suppressed the accumulation of total cholesterol. Besides, dietary lycopene increased mRNA expressions of critical genes related to the reverse cholesterol transport (RCT) in the lung [73]. Similarly, lycopene is protective against NNK/CS induced lung lesions and pulmonary cholesterol homeostasis through the restoration of the impaired RCT [74]. Lycopene supplementation could inhibit α7 nicotinic acetylcholine receptor’s expression in NNK-induced mice. Moreover, it attenuated the mortality and pathological lesions in the lung [75]. The lycopene was converted to apo-10′-lycopenoids, which dose-dependently decreased the lung tumor multiplicity in NNK-induced A/J mouse model [76]. Similarly, anti-carcinogenic and antioxidant functions of apo-10′-lycopenoids were mediated by inducing phase II detoxifying/antioxidant enzymes as well as activating the expression of Nrf2 [111]. Apo-14′-lycopenoic acid is the lycopene eccentric cleavage product. Apo-10′-lycopenoic acid and apo-14′-lycopenoic acid decreased cigarette smoke extract-induced ROS production, 8-OH-dGuo formation and COX-2 expressions [112]. All these nicotine-induced structural changes in lungs of offspring mice were prevented by supplementing the mother’s diet with tomato juice [113]. Tomatoes can be eaten as either fruit or vegetable, and based on the current literature, there seems to be no risk of toxicity when consumed in moderation. In a word, tomato as a common fruit offers health benefits in daily life.

#### 2.3.3. Grape

The grape contains various phenolic compounds, flavonoids and stilbenes. Many components in grapes have chemopreventive effects on lung cancer. For example, resveratrol is an important phenolic component in grapes. It has been reported that resveratrol exhibited a 27% reduction in tumor multiplicity, resulting in a 45% decrease in tumor per mice in NNK-induced A/J mice [77]. Resveratrol also prevented BaP-induced CYP1A1 expression [78]. Ellagic acid (EA) is produced by hydrolysis of ellagitannins, which is present in fruit berries and grapes. It has been reported that EA decreased ATP levels, reduced HIF-1α and activated AMP-activated protein kinase (AMPK) in lung tissues [114]. EA down-regulated the level of cancerous inhibitor of protein phosphatase 2A (CIP2A) in tumor-bearing mice [115]. 2,6-dimethoxy-1,4-benzoquinone (DBQ) was isolated from *Vitis coignetiae* (crimson glory vine, also named yamabudo in Japan). Yamabudo-juice or DBQ showed chemopreventive effects on lung tumors in NNK-induced mice. Investigation of the anti-tumorigenic mechanisms of yamabudo juice and DBQ indicated they reduced methyl DNA adducts and accelerated DNA repair. In addition, yamabudo juice and DBQ could suppress the signaling of Akt, Erk and Stat3 [79]. Grape seed proanthocyanidins (GSPs) are promising phytochemicals that have anti-inflammatory and anti-cancer effects [116]. Researchers proved GSPs inhibited the growth of lung tumor xenografts in nude mice; this effect was connected with the reduction of Bax, capase-3 [80]. Moreover, GSPs also reduced radiation damage to normal lung tissues and also regulated the expression of Ki-76, p53,Il-6 and TNF-α [117]. In short, grapes can be more used in life for whatever aim. Grapes are a common fruit and are also used as raw materials for wine. Grapes did not seem to have much of a protective effect against lung cancer, which may be partly due to the way resveratrol was administered. As a fruit, however, there seems to be no harm in eating more of it in daily life. 

### 2.4. Vegetables 

#### 2.4.1. Cruciferous Vegetables

##### Phenethyl Isothiocyanate 

Cruciferous vegetables have been traditionally consumed in the human diet as fresh and preserved vegetables, vegetable oils, and condiments since ancient times. 2-Phenethyl isothiocyanate (PEITC) is a natural product found in cruciferous vegetables. Various biological activities of PEITC have been explored. PEITC can induce detoxifying enzymes such as GST. Additionally, it can inhibit CYP450 involved in metabolic activation of carcinogens [118]. In the matter of lung cancer prevention, treatment with PEITC inhibited lung carcinogenesis in NNK-exposed mice [119,120]. PEITC could inhibit NNK-induced lung tumors by suppressing metabolic activation of NNK and inhibiting α-hydroxylation of NNAL in lung tissues of male F344 rats [81]. Moreover, dietary PEITC decreased the incidence of alveolar atypical and the expression of COX-2 and PCNA [82]. In a clinical trial, the NNK metabolic activation ratio was reduced after PEITC treatment in 82 smokers [102]. Similarly, Boldry et al. reported the effect of PEITC on the metabolism of carcinogen 1,3-butadiene in current smokers. The results revealed PEITC treatment is protective in respect of the detoxification of 1,3-butadiene in cigarette smokers [103]. Other compounds, including sulforaphane and sulforaphane-N-acetylcysteine, have been documented the chemopreventive efficacy on lung cancer. Dietary sulforaphane-N-acetylcysteine significantly reduced the incidence of lung tumors. Furthermore, the malignant lung tumor multiplicity was significantly reduced with dietary sulforaphane and sulforaphane-N-acetylcysteine in NNK-induced A/J mice by decreasing PCNA and capase-3 [121]. 

##### Indole-3-Carbinol

Indole-3-carbinol (I3C) is a constituent in Brassica vegetables. I3C is reported to reduce the tumor multiplicity and the level of tumor-associated signature proteins, such as apolipoprotein A-1, a-antitrypsin precursor and transferrin [83]. In NNK induced A/J mice, I3C decreased tumor burden and size by inhibiting PI3K/Akt signaling [84]. Given dietary I3C, in NNK plus BaP-treated mice, I3C showed a dose-dependent inhibition of lung tumor per mouse. Furthermore, I3C restrained the expression of PCNA, p-Akt as well as the number of Ki-67–positive cells [85]. Moreover, I3C has also been reported in combination with other compounds to prevent lung cancer. For example, a given combination of PEITC-N-acetylcysteine (NAC) and I3C could effectively reduce tumor multiplicity in NNK plus BaP induced A/J mice. The combination treatments could reduce the activation of Akt and NF-kB in lung tumor tissues that were pretreated with cigarette smoke (CSC) condensate [122]. In addition, previous research indicates that the mixtures of I3C and silibinin are much stronger than the individual compounds in A/J mice [123]. The cruciferous plant family is one of the most studied in the field of lung cancer prevention. However, it seems to have no effect on cancer treatment. It may be used in combination with chemoradiotherapy drugs in the future to reduce the side effects of chemoradiotherapy drugs to a certain extent.

#### 2.4.2. Cucurbitaceae

##### Cucurbitacin B

The major crop species of cucurbitaceae family are cucumber, melon and watermelon. The chemical constituents of cucurbitaceae mainly include terpenoids, glycosides, alkaloids, saponins, tannins, etc. Cucurbitacin B (CuB) is one of the major terpenoids in cucurbitaceae plant. CuB could suppress both DNMTs and histone deacetylase as well as inhibit tumor incidence and multiplicity in A/J mice [86]. Meanwhile, Hua et al. confirmed the protective effect of CuB on lung function. CuB effectively improved the pulmonary gas exchange function, reduced pulmonary edema, and inhibited the inflammatory response by reducing the level of TNF-α and IL-6 in lung tissues [124].

### 2.5. Spices

#### 2.5.1. Garlic

Garlic (*Allium sativum* L.), belonging to genus allium in the Liliaceae family, is native to central Asia and is widely cultivated around the world. It has been used to heal various diseases, including leprosy, cancer, constipation, asthma, and fever. Garlic has abundant bioactive chemical compounds, such as diallyl sulfide (DAS), diallyl disulfide (DADS) and diallyl trisulfide (DATS). All of these compounds have been reported to have the effects of anti-carcinogenesis [125]. Several studies have revealed the chemoprevention of DAS in lung cancer. Hong et al. found that DAS significantly lowered the incidence of lung tumors and the tumor multiplicity in A/J mice that carcinogens induced. DAS also inhibited the metabolism of NNK in lung microsomes [87]. Oral DAS also decreased tumor marker enzymes and recovered antioxidant enzymes, SOD and CAT in mice [90]. Investigation by Kong et al. revealed that oral DAS partially could reverse parts of the mitochondrial metabolic pathways, global methylation and transcriptomic changes [88]. Moreover, DAS inhibited CYP enzymes activation for inhibiting the bioactivation of NNK in A/J mice of lung tissues [89]. DAS inhibited pulmonary adenoma formation and increased glutathione S-transferase [126]. DATS and DADS induced apoptosis by increasing DNA fragmentation and activating C-Jun N-terminal kinase (JNK), up-regulating p53, and down-regulating Bcl-2 expression [127]. In DATS treated lung cancer cells it was found that Bax and Bak proteins are critical targets [128]. Garlic oil is thought to be bioactive in garlic. Our group recently screened the inhibitory effects of garlic oil on NNK treated A/J mice. The results displayed that garlic oil increases the level of the phase II drug-metabolizing enzymes for inhibiting the NNK-induced lung cancer [7]. 

#### 2.5.2. Chilli

Capsaicin, a component of red chilli and red pepper, has been found to have various effects, including anti-cancer. Capsaicin could inhibit the process of lung carcinogenesis by inducing apoptosis, increasing DNA fragmentation, up-regulating the expressions of p53, Bax and decreasing the level of Bcl-2 in BaP-induced in Swiss albino mice [92]. Capsaicin also reversed BaP-induced increase of extracellular matrix components and proteases [93]. Capsaicin induced apoptosis in lung cancer cells was mediated via the TRPV6 receptor as well as inversed oxidative stress in mice administered with BaP [94,129]. Moreover, capsaicin decreased lung mitochondrial lipid peroxidation in Bap-induced lung cancer Swiss albino mice. More importantly, capsaicin augmented the activities of enzymic and non-enzymic antioxidants, citric acid cycle enzymes and respiratory chain enzymes to near normalcy [130]. Previous studies have also found that capsaicin supplementation in lung cancer bearing mice considerably prevented the increaseof TNF-α, IL-6, COX-2 and NF-κB [91].

#### 2.5.3. Zingiber

Ginger, the rhizome of Zingiber officinalis, is one of the most widely used species in Asia. Phytochemical studies revealed that ginger is rich in chemicals including 6-shogaol, β-Elemene, Zerumbone and so on. Zerumbone, presented in subtropical ginger Zingiber zerumbet Smith, is reported to have anti-cancer properties. In the process of lung cancer induced by NNK, administration of Zerumbone restrained the multiplicity of lung adenomas through inhibition of proliferation and induction of apoptosis. These effects mainly resulted from suppression of NF-κB, HO-1 expression in lung tissues [95]. 6-shogaol is a major component of dry ginger. It increases cAMP concentration, limits NF-κB signaling and produces pro-inflammatory cytokines in activated CD4 cells to attenuate house dust mite antigen mediated lung inflammation in mice [131]. 

#### 2.5.4. Turmeric

Curcumin is a constituent present in *Curcuma longa* L. (turmeric) rhizome. It has been used as a spice for cooking in Asia for many years. It has been reported that curcumin has chemopreventive effect on lung cancer. The average number of tumor nodules present in the lungs of the Swiss albino mice induced by BaP was significantly higher than in mice that received both a 2% curcumin diet and BaP. Furthermore, a curcumin diet lowered BaP-induced activation of NF-κB, MAPK signaling and COX-2 transcription in lung tissues of A/J mice [96]. Similarly, dietary curcumin reduced BaP-induced DNA adduct, oxidative damage and inflammation [132]. In a word, a series of spices such as garlic, ginger, pepper and turmeric is present in daily diets and living habits, but their smell is strong and will be disliked by some people. Moreover, it is not certain whether excessive consumption will have toxicity. 

### 2.6. Other Plants

Flax has a long history of use as a food, medicine, and textile fiber. Chikara et al. found that flaxseed consumption could protect against both NNK and BaP induced lung tumor formation in A/J mice lung tissues. The effects mainly include upregulating the expression of phase II enzymes and downregulating the expression of proinflammatory cytokines [97]. *Houttuynia cordata* Thunb (HCT) is a commonly used herbal medicine in Asian countries, with anti-inflammatory and anti-bacterial effects. In BaP-induced A/J mice, HCT extracts activated the Nrf2-HO-1/NQO-1 signaling pathway. It also has the function of relieving intracellular ROS generation to attenuate DNA damage and inflammation induced by BaP stimulation [98]. *Camptosorus sibiricus* Rupr (CSR) is a commonly used herbal medicine which has anti-tumor effects. CSR extracts prevented lung tumorigenesis by acting against ROS and DNA damage in BaP-induced A/J mice [99]. Fermented brown rice and rice bran have chemopreventive effects on carcinogenesis in the colon, liver and stomach. For lung cancer prevention, both fermented brown rice and rice bran inhibited NNK-induced pulmonary tumorigenesis in A/J mice. This effect is mainly through reducing the expression of mRNA levels of CYP4502A5 [100].

## 3. Future Directions

Overall, this review highlights phytochemicals as potential chemopreventive agents against lung cancer. Accumulating evidence reveals that many dietary natural resources and their bioactive ingredients are able to prevent tobacco related lung cancer. Mechanically, dietary phytochemicals mainly achieved the chemoprevention of tobacco carcinogens induced lung tumorigenesis by regulating phase II metabolizing enzymes and reducing DNA adducts as summarized in Figure 2. Additionally, dietary phytochemicals are able to target and reverse epigenetic changes caused by environmental exposure, which may be due to their high potency, lower toxicity, safety, and cost-effectiveness. It is promising that the development of treatments based on the consumption of natural phytochemicals for cancer chemoprevention will produce more positive outcomes in the future, and offer the possibility of reducing cancer risk in society. An increasing number of studies have demonstrated that a large number of dietary phytochemicals can be used alone or in combination with traditional chemotherapeutic drugs to prevent the cancer initiation and metastatic spread. Rational usage of dietary phytochemicals may reduce the side effects of chemotherapy drugs. Therefore, dietary phytochemicals could be useful supplements to enhance chemotherapeutic outcomes and benefit lung cancer patients, especially providing a lung cancer prevention strategy for people at risk of developing cancer, both those who smoke and those exposed to second-hand smoke over the long term. However, it is essential to explore the potential mechanisms of action of various dietary components in combination with chemotherapies. 

In addition to the advantage in lung cancer prevention, dietary chemopreventive phytochemicals still have some bottlenecks. Up to now, PEITC and Kava have been used in clinical experiments, but the results are not satisfactory, and the role in lung cancer prevention has not achieved the expected results. Potential chemopreventive phytochemicals have not yet shown significant clinical effects in preventing lung cancer progression in humans. Moreover, high concentrations of phytochemicals for long-term use may be toxic to humans. Note must be taken of biological differences, metabolic conversion, transport mechanisms, tissue availability in the metabolization of the compounds, and their bioavailability. Therefore, it is too early to draw conclusions about their chemoprevention effect. It is plausible to develop large-scale, randomized, double-blinded, placebo-controlled clinical trials before their use in the treatment of cancer. In the future, the chemopreventive efficacy of dietary phytochemicals on lung cancer should be further investigated. 

## Figures and Tables

**Figure 1 nutrients-15-00491-f001:**
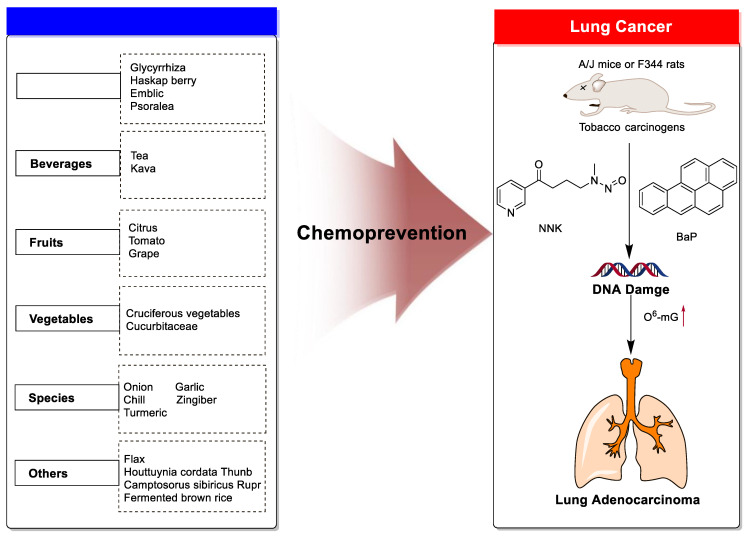
Graphic summary of the dietary phytochemicals that are cancer-preventive on BaP or NNK triggered lung carcinogenesis.

**Figure 2 nutrients-15-00491-f002:**
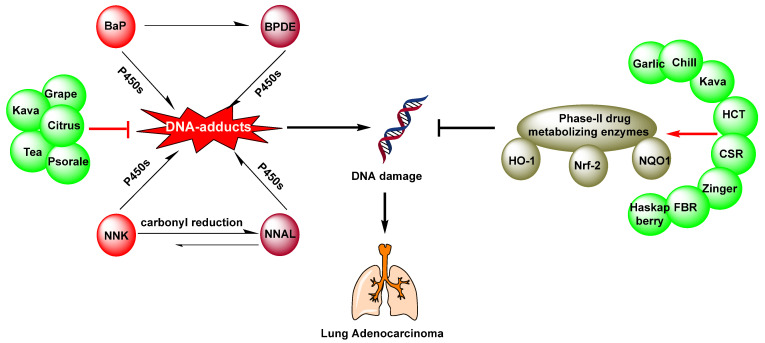
The mechanism of phytochemical prevention of tobacco-induced lung cancer.

**Table 1 nutrients-15-00491-t001:** Summary of the in vivo studies describing dietary phytochemicals for lung cancer prevention.

Nature Products	Bioactive Components	Classification of Compounds	Dose	Model of Animals	Effects and Mechanisms	Ref
Glycyrrhiza	LicA	Monomeric compound	20 mg/Kg	NNK-induced A/J mice	Reversed NNK-induced miRNA expression	[29]
	Glycyrrhiza	Monomeric compound	100 mg/Kg	PDX mice	Inhibited the growth of lung tumor; Suppressed migration and invasion; Decreased the activity of Jak/KSTAT signaling	[30]
	Glycyrrhiza + cisplatin	Monomeric compound	135 mg/Kg + 2.5 mg/Kg 50 mg/Kg/2 d	Nude mice	Reduced expression of thromboxane synthase (TxAS) and PCNA; Rescued damage of liver and kidney	[31,32]
Haskap berry	Cyanidin-3-O-glucoside	Monomeric compound	0.2 g/mice/d	NNK-induced A/J mice	Reduced tumor multiplicity, tumor area, PCNA and Ki-67	[33]
	Cyanidin-3-O-glucoside plus 5-FU	Monomeric compound	5 mg/Kg + 25 mg/Kg	Nude mice	Decreased inflammatory cytokine levels, such as IL-1β,IL-6 and TNF-α; Decreased COX-2, NF-kpa, PCNA	[34]
Emblic	EmblicL extracts	Mixture	5 g/kg	BaP-induced A/J mice	Reduced the number of nodes by regulated the IL-1β/miR-i101/Lin28B signaling pathway	[35]
Psoralea	8-MOP	Monomeric compound	50/12.5 mg/Kg	NNK-induced A/J mice	Inhibited the CYP2A5-mediated metabolic activation of NNK in the mouse lung; Reduced NNK-induced tumor incidence and tumor multiplicity	[36,37,38]
			12.5 mg/Kg	NNK-induced C57BL/C mice	Decreased α-hydroxylation of NNK	[39]
Tea	EGCG	Monomeric compound	0.5% diet	NNK-induced A/J mice	Attenuated DNMT1 and inhibited lung tumorigenesis	[40]
	Caffeine	Monomeric compound	0.044% in drinking water	NNK-induced A/J mice	Inhibited the progression of lung adenoma to adenocarcinoma	[41]
			680 ppm in drinking water	NNK-induced F344 rats	Decreased the incidence of lung tumor	[42]
	PBPs	Mixture	0.3% in drinking water	NNK-induced A/J mice	Inhibited the progression of adenoma to adenocarcinoma and cell proliferation	[43]
			1.5% in drinking water	NNK- and BaP-induced A/J mice	Inhibited inflammation and induced apoptosis via p38 and Akt	[44]
			3% in drinking water		Anti-initiating and anti-promoting and inhibited the expression of CYP1A2 and increased the activity of GSTs and decreased tumor incidence and multiplicity	[45]
	Polyphenon E	Mixture	0.5% in drinking water	NNK-induced A/J mice	Reduced the incidence and multiplicity; Inhibited cell proliferation and enhanced apoptosis and lowered levels of c-Jun and extracellular signal-regulated kinase (Erk) ½ phosphorylation	[41]
	Polyohenon E plus Eatorvastatin	Mixture	0.5% in drinking water + 200 ppm diet	NNK-induced A/J mice	Suppressed myeloid cell leukemia 1; Increasing the cleaved caspase-3 and cleaved poly (ADP)-ribose polymerase level	[46]
	Catechin hydrate	Mixture	20 mg/Kg	BaP-induced albino rats	Reduced Bcl-2, Bax and Capase-3 expression	[47]
	Tea preparation	Mixture	0.6%, 2%, 0.3% in drinking water	NNK-induced A/J mice	Reduced lung tumor multiplicity; Inhibited angiogenesis and enhanced apoptosis index; Reduced the number of lung tumors; Inhibited the increase of 8-OH-dGuo level; Inhibited the percentage of PD-L1	[48,49,50]
			1% in drinking water	Bap-induced SD rats	Decreased the level of ROS, LPO and NO	[51]
Kava	Kava	Mixture	5 mg/g 10 mg/g	NNK-induced A/J mice	Reduced tumor multiplicities; Decreased O^6^-mG; Suppressed NF-κB; Enhanced apoptosis	[52,53,54]
			10 mg/g	NNK plus Bap-induced A/J mice	Reduced PCNA; Increase in caspase-3, and cleavage of poly (ADP-ribose) polymerase (PARP); Inhibited the activation of nuclear factor κBNF-κB	[55,56]
			500 mg/Kg	Acetaminophen induced C57BL/6 mice	Increased AST and ALT; Increased severity of liver lesions	[57]
	DHM	Monomeric compound	2 mg/d 32 mg/Kg 1 mg/g 0.05 mg/g 200 ppm diet	NNK-induced A/J mice	Decreased the formation of O^6^-mG; Increased the urinary excretion of NNK metabolites; Increased NNAL-O-glue; Decreased adenoma multiplicity; Reduced DNA adducts; Suppressed PKA pathway	[58,59,60,61,62,63]
			1 mg/g	NNK-induced C57BL/6 mice	Reduced the formation of O^6^-mG adducts	[64]
Citrus	5-Demethylnobiletin	Monomeric compound	0.05% diet	NNK-induced A/J mice	Decreased lung tumor multiplicity and tumor volume	[65]
	Nobiletin	Monomeric compound	500 ppm diet	NNK-induced gpt delte mice	Inhibited CYP enzymes that involved in the metabolic activation of NNK	[66]
			0.05% diet	NNK-induced A/J mice	Decreased tumor volume; Decreased the expression level of PCNA	[67]
	Limonin	Monomeric compound	50 mg/Kg	Bap-induced A/J mice	Decreased lipid peroxidation, serum marker enzymes and inflammatory cytokines levels; Enhanced apoptosis	[68]
	β-Cryptoxanthin	Monomeric compound	10 mg/Kg	NNK-induced A/J mice	Down-regulated α7-nAChR/PI3K signaling	[69]
			20 mg/Kg	NNK-induced A/J mice	Restored the nicotine-suppressed expression of lung SIRT1, p53, and RAR-β; Decreased the levels of lung il-6 mRNA and phosphorylation of AKT	[70]
	Hesperetin/Naringenin /Hesperidin	Monomeric compound	25 mg/Kg 50 mg/Kg	BaP-induced Swiss albino mice	Alleviated LPO, modulated antioxidants and decreased the expression of NF-kB, PCNA and CYP1A1; Attenuated mitochondrial dysfunction	[71,72]
Tomato	Lycopene	Monomeric compound	6.6 mg/Kg	NNK/CS-induced ferrets	Inhibited chronic bronchitis, emphysema, and preneoplastic lesions; Increased mRNA expression of critical genes related to RCT in the lungs	[73] [74]
			6.6 mg/Kg	NNK-induced ferrets	Prevented the expression of α7 nicotinic acetylcholine receptor in the lung; Decreased the mortality rate of ferrets	[75]
	apo-10′-lycopenoids	Monomeric compound	120 mg/Kg	NNK-induced A/J mice	Decreased lung tumor multiplicity	[76]
Grape	Resveratrol	Monomeric compound	60 mg/Kg	NNK-induced A/J mice	Inhibited the level of CYP450	[77]
			50 mg/Kg	BaP-induced Balb/C mice	Prevented CYP 1A1 expression	[78]
	DBQ	Monomeric compound	0.1 g/L drinking water	NNK-induced A/J mice	Decreased lipid peroxidation	[79]
	Grape seed proanthocyanidins	Mixture	0.5% diet	Nude mice	Reduce Bax, capase-3	[80]
Cruciferous vegetables	PEITC	Monomeric compound	3 μMol/g diet 0.5 g/Kg diet	NNK-induced F344/Wistar rats	Inhibited metabolic activation of NNK; inhibited lung α-hydroxylation of NNAL; Decreased COX-2 expression and PCNA expression	[81,82]
	I3C	Monomeric compound	112 mM/g diet 10 μM/g diet	NNK-induced A/J mice	Reduced tumor multiplicity; Reduced the level of tumor-associated signature proteins; Inhibited PI3K/Akt signaling	[83,84]
			112 mM/g diet	NNK plus BaP-induced mice	Reduced the number of Ki-67–positive cells and expression of proliferating cell nuclear antigen, phospho-Akt, and phospho-BAD	[85]
	Cucurbitacin B	Monomeric compound	0.2 mg/Kg	NNK-induced A/J mice	Inhibited tumor incidence and multiplicity; Inhibited DNA methyltransferase (DNMTs) and histone deacetylase (HDACs)	[86]
Garlic	DAS	Monomeric compound	200 mg/Kg	NNK-induced A/J mice	Decreased the incidence of lung tumors and multiplicity; inhibited the metabolism of NNK in mouse lung microsomes Reversed mitochondrial metabolic pathways, global methylation and transcriptomic changes Inhibited the bioactivation of NNK by inhibition of other CYP enzymes active	[87,88,89]
			100 mg/Kg	BaP-induced Swiss mice	Decreased tumor marker enzymes and recovered antioxidant enzymes, SOD and CAT	[90]
	Garlic oil	Mixture	50 mg/Kg	NNK-induced A/J mice	Induced the expressions of the phase II drug-metabolizing enzymes, and HO-1	[7]
Chilli	Capsaicin	Monomeric compound	10 mg/Kg	BaP treated A/J mice	Prevented the increasing of TNF-α, IL-6, COX-2 and NF-κB	[91]
			10 mg/Kg	BaP-induced Swiss albino mice	Inhibited the development lung carcinogenesis; Induced apoptosis, through inducing increase DNA fragmentation and the expressions of p53, Bax and caspase-3 and decreasing the level of Bcl-2	[92] [93]
			10 mg/Kg		Inversed BaP-induced oxidative stress	[94]
Zinger	Zerumbone	Monomeric compound	500 ppm diet	NNK-induced A/J mice	Inhibited the multiplicity of lung adenomas Induced apoptosis, and suppressed NF-κB and HO-1 expression	[95]
Turmeric	Curcumin	Monomeric compound	2% diet	BaP-induced Swiss albino mice	Decreased lung nodes; Reduced the activation of NF-κB and MAPK signaling and Cox-2 transcription	[96]
Flax	Flaxseed	Mixture	10% diet	NNK-induced A/J mice	Reduced lung tumor incidence and multiplicity; Suppressed the phosphorylation (activation) of p-AKT, p-ERK, and p-JNK kinases	[97]
HCT	HCT extracts	Mixture	50 g/Kg	BaP-induced A/J mice	Activated the Nrf2-HO-1/NQO-1 signaling pathway intracellular ROS generation, attenuated DNA damage and inflammation	[98]
CSR	CSR extracts	Mixture	3 g/Kg	BaP-induced A/J mice	Suppressed ROS production by re-activating Nrf2-mediated reductases HO-1 and NQO-1	[99]
FBRA	FBR	Mixture	10% diet	NNK-induced A/J mice	Decreased the expression of CYP-450 and Ki-67 positivity	[100]

**Table 2 nutrients-15-00491-t002:** Summary of the clinical trials describing dietary phytochemicals for lung cancer prevention.

Nature Products	Bioactive Components	Classification of Compounds	Dose	Effects and Mechanisms	Ref
Kava	Kava	Mixture	225 mg/d	Increased urinary excretion of total NNAL; Reduced 3-mA	[101]
Cruciferous vegetables	PEITC	Monomeric compound	10 mg dissolved in 1 ml olive oil 4 times/d, lasts 5 days	Reduced NNK metabolic activation ratio; Increased urinary levels of BD-mercapturic acids	[102,103]

**Table 3 nutrients-15-00491-t003:** The representative dietary plants and phytochemicals for lung cancer prevention.

Plants	The Structure of Nature Products
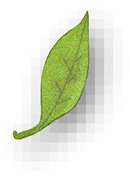 Tea	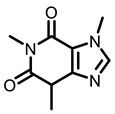 Caffeine	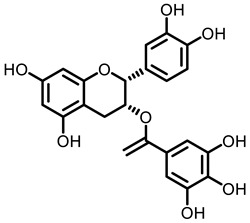 EGCG
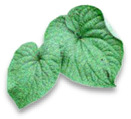 Kava	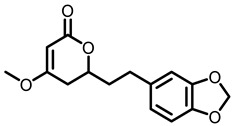 Dihydromethysticin
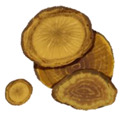	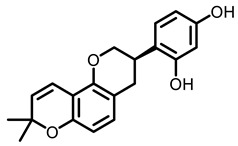 LicA	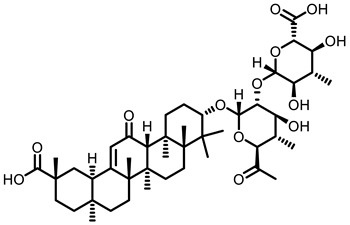 Glycyrrhizin
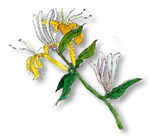	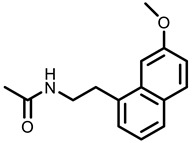 C3G
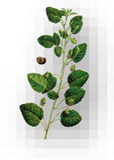 Psoralea corylifolia L.	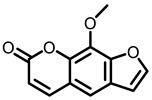 8-MOP
Grape 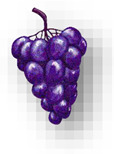	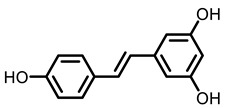 Resveratrol	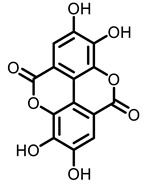 EA	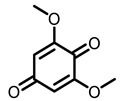 DBQ
Citrus 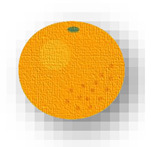	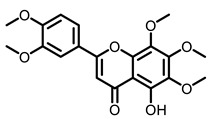 5-Demethylnobiletin	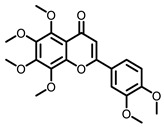 Nobiletin	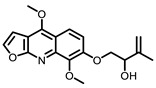 Limonin	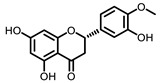 Hesperetin
	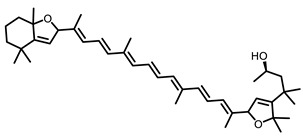 BCX	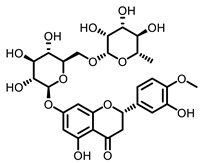 Hesperidin	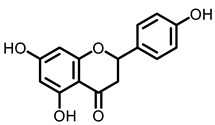 Naringenin
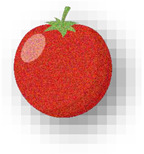 Tomato	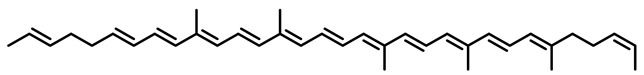 Lycopene
Cruciferous vegetables 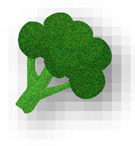	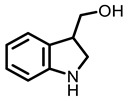 I3C	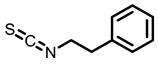 PEITC
Cucurbitaceae 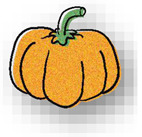	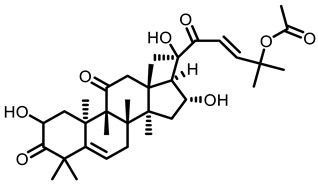 Cucurbitacin B
Garlic 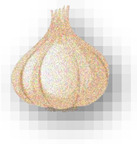	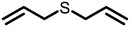 DAS	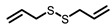 DADS	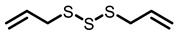 DATS
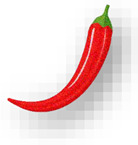 Chilli	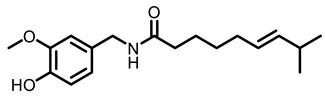 Capsaicin
Ginger 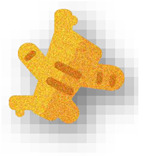	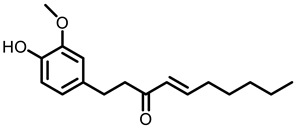 6-shogaol	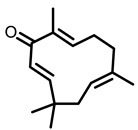 Zerumbone
Turmeric 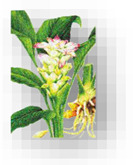	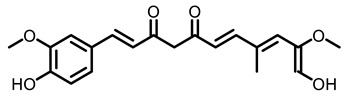 Curcumin

## Data Availability

The data in this study are available upon reasonable upon reasonable request.

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
