# Peer review of "Dietary Phytochemicals as Potential Chemopreventive Agents against Tobacco-Induced Lung Carcinogenesis"

_nutrients, 2023, doi:10.3390/nu15030491_

Round 1
Reviewer 1 Report (Previous Reviewer 1)
The manuscript has been improved over the previous version albeit still with some flaws. First of all, it is too superficial. Carcinogenesis is a multistep process consists of initiation, promotion and propagation steps. The article lacks information on which stages of carcinogenesis phytochemicals may exert their chemopreventive effects. The scientific value of the article is also diminished by the inaccurate concept adopted by its authors, who focused only on in vitro studies/mouse model, completely ignoring the clinical aspect/studies. The in vitro studies are conducted using ridiculous concentrations of compounds that have no relation to the realistic levels achievable from the consumption of plant-derived foods. I absolutely disagree with the statement in line 506. There are no clinical studies of any kind to support the authors' claim about the supportive role of compounds in lung cancer chemotherapy. Moreover, the authors do not cite any epidemiological studies that would support another claim made in the second part of this sentence (regarding lung cancer prevention). Such claims in scientific texts only introduce chaos and confusion and contribute to the development of all kinds of pseudo-scientific theories.
Author Response
See attached file.

Reviewer 2 Report (Previous Reviewer 2)
· Line 28-30: Why did the authors focus on the US and China? Please include evidence on the effect of cigarette smoking/second hand smoking on the risk of LC mortality among men and women worldwide. I would suggest referring to this reference: Institute for Health Metrics and Evaluation. Global Burden of Disease 2017. 2017. Available online: http://vizhub.healthdata.org/gbd-compare/#
· Line 74: It would be benefit to expand on the role of α7 nicotine acetylcholine receptor (α7nAChR) in LC development. Activation of α7nAChR promotes cell proliferation, metastasis, angiogenesis, and reduce apoptosis in LC cells. Please refer to this article which discusses this point (Int. J. Environ. Res. Public Health 2021, 18(10), 5243).
· Line 109-111: There is no clear rationale here. I want to see 2-3 sentences explaining why this review is important? What this review adds in light of previous reviews?
· Line 112-116: This should be in a separate section “Methods”. Also, the search terms are unclear-please rephrasing. I suggest the following “Phytochemicals” OR "fruits and vegetables" OR "beverages" OR "species" AND “Lung cancer” AND "cigarette smoking" AND "NNK" OR “BaP” OR "nicotine". The search is limited to in vivo studies.
· Line 117: It is unclear to me why Table 1 is presented here. It should be before the conclusions after discussing all the sections.
· Table 1: Could the authors please include a column that shows the types of LC cells affected by phytochemicals (e.g., A549, H1299, H460, DMEM…..etc). This should also be clearly mentioned within the text starting from Line 136.
· Table 1: In the dose column, concentrations in some studies should be clarified; e.g., 0.5%, 0.044%, 0.3% per body weight.
Round 2
Reviewer 1 Report (Previous Reviewer 1)
Information about conclusions from AACR CANCER PROGRESS REPORT 2022 must be included. Also, description of epidemiological investigations, about eating at least 30 g or more vegetables and fruits should be provided.
Author Response
See attached file.

Reviewer 2 Report (Previous Reviewer 2)
No further comments.
Author Response
Thanks again for your review.
This manuscript is a resubmission of an earlier submission. The following is a list of the peer review reports and author responses from that submission.
Round 1
Reviewer 1 Report
In the present work, the authors report an review focused on Dietary phytochemicals as the potential chemopreventive agents against lung cancer (LC). It is the interesting topic as LC is regarded as increasing medical problem in Western and East countries, mostly due to tobacco usage. In my view, the manuscript is within the scope of Nutrients, however the quality of the paper is not adequate. The text needs major revision as follows: The manuscript is too vague. For example authors stated that “BaP can induce DNA break”. It's a generalization – DNA breaks could arise when DNA or RNA Polymerase meets DNA adduct. Tables are also not clear. Authors use many abbreviations without further explanations like FBRA, HCT, CSR, A/J mice, PDX mice and so on. I suggest add one more manuscript part between 1 and 2 focused on detailed mechanism of BaP and NNK induced cancerogenesis as well as mouse models used for chemoprevention of lung cancer studies in the context of the information provided in section 2 and Tables (for example what is the ectopic miRNA induced by NNK that are reversed by lipochalcone A?). The text of section is tedious. The authors are recommended to summarize all sub-section succinctly. The author lists many factors and describes their function and characteristics, however, there is only limited comparison or discussion of them. I would suggest to add more discussion rather than repeat information reported by others.
Reviewer 2 Report
· Line 13-14: Suggest “This review aims to summarize the published studies in the past 20 years exploring the dietary phytochemicals............ using the PubMed, Web of science, Scopus databases”.
· Line 28: Please clarify here. “Lung cancer is divided into two broad histologic classes: small-cell lung carcinomas (SCLC) and non-small lung cancer (NSCLC). NSCLC is further classified into three phenotypes, including adenocarcinoma........
· Line 29-42: Much progress have been achieved during last decade including combination treatments with chemo+ IO and targeted therapies for mutated patients like EGFR/ALK/ROS1/KRAS pts. Should put more focus on the progress during last years, and despite that to mention that the prognosis still dismal. Also, smoking increases lung cancer but not all types of lung cancer. For example, some lung cancer patients harboring EGFR mutations do not smoke. This should be clearly highlighted.
· Line 39-40: What about nicotine? Although it is not carcinogenic, it induces NNK.
· In introduction, authors do not have the strong-enough biological background to review the molecular biology of lung cancer. Also, it would be interested if the authors state the molecular role of medicinal plants with the variety of pathways affected in NSCLC lines. I would recommend referring to these interesting articles (Int. J. Environ. Res. Public Health 2021, 18(10), 5243, Molecules. 2020 Jan 6;25(1):231; Evid Based Complement Alternat Med. 2014;2014:604152).
· Line 71-74: This should be clarified; e.g. what type of vegetables? Why fruits are potential protective effects on lung cancer development in non-smokers only?
· Authors should clearly mention the effects of combination of medicinal plant extracts with chemotherapeutics/radiotherapy in smoking-induced lung cancer treatment. There are many studies conducted in this area. For example, Int J Mol Sci. 2022 Jul 18;23(14):7905.
· The novelty of this review is very weak. The last paragraph in introduction could be improved. I would suggest that the authors to present the aim of the paper with regards to what is currently known by in vivo/vitro studies, therefore highlighting the added value of this study.
· The method section is missing. Please describe the search terms and types of study designs (e.g., in vivo, in vitro, RCT..…etc.) you used, the databases (e.g., PubMed) you searched, the inclusion/exclusion you addressed....etc.
· Table 1 should be clearly organized. The table could have columns such as study type (in vivo/in vitro), lung cancer cell type, extracts, dosage, anti-lung cancer activity…etc). the table should be moved after section 2.
· Some discussions on the doses of bioactive compound used in various studies compared to their extended dosage in vivo would be helpful to understand the effects due to the right dosage and overdosage on cells. It could be interesting to suggest a dosage to support chemotherapy, distinguish if cytotoxic or not: and for a prevention purpose.
· A paragraph should be devoted to bioavailability, as some of the studies cited are in vitro.
· Refs 83-86, 102 are very old- Please update.
· English language should be improved in this paper.
Reviewer 3 Report
Abstract
Line 13 is missing a verb: protect, reduce risk?
Introduction
Line 67 it is a very strong sentence to say that treatments kill , probably we can say that the immunosuppression promoted and the effects of cancer itself lead to low life expectancy
And the reference is not adequate for this
Line 70
You can introduce the subject referring that diet and specific dietary compounds may have a protective effect in cancer, including lung cancer , and refer to food groups that are mostly associated with this effect
In order to improve manuscript organization I would suggest you introduce the main mechanisms how dietary phytochemicals can exert protective effects and prepare a table with the studies that report this instead of table 2. Report: type of study, compounds/food/ingredient, design , main result/outcome. This will improve manuscript understanding because it is very dense and requires a clearer approach.
Round 2
Reviewer 1 Report
The manuscript has been improved sine last time.
Author Response
Response: Thanks again for your review.
Reviewer 2 Report
Dear Authors,
The paper has significantly improved by these revisions. Two points remain:
1. Table 1: Given this review focused on in vivo studies, I would suggest deleting the study type column. The title of table should be "Summary of the in vivo studies describing dietary phytochemicals for lung cancer prevention".
2. Please double check the reference list. Some references are duplicated (e.g., Ref # 24 & 79).
Author Response
1. Table 1: Given this review focused on in vivo studies, I would suggest deleting the study type column. The title of table should be "Summary of the in vivo studies describing dietary phytochemicals for lung cancer prevention".
Response: We have deleted the study type column of the table and changed the title of table as suggested.
2. Please double check the reference list. Some references are duplicated (e.g., Ref # 24 & 79).
Response: We have carefully checked all the references to ensure the citations are correct.
Reviewer 3 Report
The manuscript significantly improved since the last version.
Author Response
Response: Thanks again for your review.